# Hierarchical Collaborated Fireworks Algorithm

**Yifeng Li and Ying Tan \***

School of Artificial Intelligence, Peking University, Beijing 100871, China; liyifeng@pku.edu.cn
* Correspondence: ytan@pku.edu.cn

**Abstract:** The fireworks algorithm (FWA) has achieved significant global optimization ability by organizing multiple simultaneous local searches. By dynamically decomposing the target problem and handling each one with a sub-population, it has presented distinct property and applicability compared with traditional evolutionary algorithms. In this paper, we extend the theoretical model of fireworks algorithm based on search space partition to obtain a hierarchical collaboration model. It maintains both multiple local fireworks for local exploitation and one global firework for overall population distribution control. The implemented hierarchical collaborated fireworks algorithm is able to combine the advantages of both classic evolutionary algorithms and fireworks algorithms. Several experiments are provided for in-depth analysis and discussion on the proposed algorithm. The effectiveness of proposed strategy is demonstrated on the benchmark test suite from CEC 2020. Experimental results validate that the hierarchical collaborated fireworks algorithm outperforms former fireworks algorithms significantly and achieves similar results compared with state-of-the-art evolutionary algorithms.

**Keywords:** fireworks algorithm; hierarchical collaboration; search space partition; swarm intelligence optimization algorithm

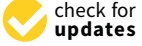

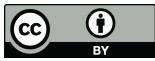

## 1. Introduction

Global optimization of non-convex problems has been a significant task for numerous academic studies and industrial applications. However, traditional gradient methods are facing difficulties in dealing with complex optimization situations, such as multi-modal or non-differentiable objective functions. Recently, a great number of evolutionary algorithms (EAs) and swarm intelligence optimization algorithms (SIOAs) have been proposed, developed, and widely applied in optimization tasks for their advantages in terms of flexibility and robustness.

The fireworks algorithm (FWA [1]) is a novel swarm intelligence optimization framework inspired from the explosion of fireworks in the night sky. Unlike the classic EAs or SIOAs, fireworks algorithm maintains several individuals called fireworks to explore different aspects of the objective function by sub-populations composed with basic individuals called sparks. Meanwhile, fireworks collaborate their local strategies to achieve efficient global optimization. For example, some studies on the fireworks algorithm have proposed variant approaches with multi-scale [2] or multi-local [3] collaboration strategies. Fireworks algorithms are particularly suitable for complex problems with a considerable number of local extrema and scenarios where large scale parallel computation is supported. Currently, it has been widely applied in fields like portfolio optimization [4], image processing [5], and power system reconfiguration [6]. It is also implemented for complex optimization scenarios like multi-objective optimization [7].

The fireworks algorithm framework is able to significantly improve the efficiency of optimization by dynamically decomposing the target problem and handling each one with a sub-population, but it can also be harmful if the decomposition and collaboration are not performed properly. For example, when optimization is decomposed into multiple local searches, collaboration must effectively integrate local information and analyze the

overall landscape of the objective function. Otherwise, information about the overall trend becomes unavailable and the efficiency of optimization is compromised.

This paper deals with this problem by combining multi-local and multi-scale decomposition and collaboration methods in the fireworks algorithm. A hierarchical model of the fireworks population is developed, analyzed, and implemented. In the proposed fireworks algorithm, as global firework controls a sub-population that approximates the overall distribution of the entire spark population, while multiple local fireworks perform local searches within dynamically partitioned sub-regions. Specifically, this paper contributes to the following aspects of fireworks algorithm.

1. **A hierarchical model.** A theoretical model based on information theory is developed for the proposed collaboration framework, which is very helpful for the design and interpretation of related algorithms.
2. **A refined control of individual search strategy.** The basic individual optimization strategy introduced from CMA-ES is analyzed and improved for the specific requirements of the FWA framework.
3. **A unified collaboration strategies.** Both multi-local collaboration and multi-scale collaboration are implemented in a unified strategy, which is intuitive but effectively combines the advantages of both approaches.
4. **An efficient FWA variant.** The proposed algorithm is able to combine the advantages of traditional EAs and FWAs. It achieves excellent results on benchmark test problems.

The remainder of this paper is organized as follows: Section 2 introduces the background of the target problem and related works. Then, Section 3 develops a theoretical model of the hierarchical collaboration framework. In Section 4, the individual optimization strategy is introduced and analyzed for global and local fireworks. The unified collaboration strategy is described in Section 5. Several sets of experimental designs and results are exhibited and discussed in Section 6. Finally, Section 7 concludes the paper.

## 2. Background

### 2.1. Problem Definition

This paper targets to solve the general continuous single-objective black-box global optimization problem with boundary constraints, which is formulated in Equation (1).

$$\mathbf{x}^* = \arg\min_{\mathbf{x} \in S} f(\mathbf{x}) \tag{1}$$

where the objective function $f : \mathbb{R}^D \to \mathbb{R}$ is sampled from an unknown distribution $p(f)$ related to specific task scenario. The search space or feasible space $S = \{\mathbf{x} \| lb_i \leq x_i \leq ub_i\}$ restricts each variable to some finite interval.

A general optimizer is used to approximates $\mathbf{x}^*$ by proposing a finite number of solutions or samples $\mathbf{x}_i$ within $S$. For the black-box problem, the objective function provides only one scalar $y = f(\mathbf{x})$ for each sample $\mathbf{x}$. A well-developed optimizer is able to combine the historical evaluations data $D_n = \{(\mathbf{x}_i, y_i)\}_{i=1}^n$ with specific prior knowledge about $p(f)$ to generate additional valid samples in each iteration.

In particular, for EAs and SIOAs, a considerable number of samples are generated in each iteration. In this case, each sample is also referred to as an individual, and the total of all individuals in each iteration (or generation) is referred to as the population. The objective function is also referred to as the sfitness function.

### 2.2. Fireworks Algorithm

Fireworks algorithm (FWA) is a novel swarm intelligence optimization algorithm that focuses on combining multiple sub-populations with a unified local optimization strategy. During the optimization process, the primary individuals called "sparks" are divided into multiple parts, each of which performs a local search for some aspect of the objective function under the leadership of an individual called "firework". The general firework algorithm performs the following steps in each optimization iteration until the

termination condition is satisfied. First, each firework distributes and evaluates explosion sparks around itself. Sometimes, several additional mutation sparks are generated utilizing the information obtained by the explosion. Then, a new generation of fireworks is selected from the current fireworks and sparks. Finally, the collaboration strategies are sometimes applied to further tune the fireworks or their explosion parameters, including the range and number of their sparks. Such a framework has significant advantages in dealing with problems with many local extremums or scenarios where a large-scale population is available. Currently, FWA has shown great potential for global optimization efficiency and has become a representative swarm-based optimization algorithm.

In the early FWAs, the individual strategies for fireworks are directly decided by comparing fitness values. The original FWA [1] decides spark number and explosion amplitude of a firework linearly to its relative fitness value. As a significant number of studies focused on the efficiency of individual search are proposed in this period, the fireworks become more and more independent. The enhanced FWA [8] made several corrections to the original FWA, including the primary explosion method. Then, both [9,10] proposed dynamic individual explosion amplitude strategies that improve FWA's efficiency significantly. Meanwhile, a large number of FWA variants with mutation operators such as [11,12] are proposed to utilize the information obtained from explosion sparks.

Then, more and more studies have spared attention to the collaboration of fireworks. The bare-bone FWA [13] achieved decent efficiency with a single firework and a minimalist strategy, which implies the failure of former collaboration methods and provides a reliable individual optimization strategy. In the cooperative framework of FWA [14], a variant algorithm with an independent selection strategy is proposed, so each firework selects a child from its sparks. A crowdedness-avoiding cooperative strategy is also applied to repel fireworks from the explosion range of the best firework. A large number of subsequent studies are based on this independent selection. In [15], a loser-out tournament strategy is proposed to estimate the potential of each firework and timely restart ones that are unlikely to outperform the current best. Recently, refs. [2,3] proposed collaboration strategies that assign fireworks to different scales or different local areas, respectively.

### 2.3. Related Works

The main contribution of this paper is to propose a hierarchical multi-population collaborative optimization framework. The related works are introduced in two directions.

#### 2.3.1. Study on Optimization Algorithm with Multiple Population

According to our knowledge, there are two primary research directions on adopting multiple populations in global optimization.

Many optimization algorithms that adopt multiple populations are called multi-population methods. In those methods, all populations evolve to solve the optimization problem simultaneously with diversified strategies or parameters. In the competition of CEC 2020 [16], several differential evolution (DE) methods achieved the best results with such a framework, including IMODE [17], J2020 [18], MP-EEH [19], and mpmL-SHADE [20]. Such methods handle the original problem with multiple sub-populations and usually collaborate by individual sharing. In contrast, the proposed algorithm handles decomposed sub-problems with sub-populations and directly collaborates strategies of each one.

Another branch of optimization algorithms with multiple populations is called co-operative co-evolution. Those methods handle large-scale problems by decomposing the variables of the solution. Ref. [21] provides an example of automatic solution decomposition and cooperative optimization. Those methods share a similar idea with the proposed algorithm but target different problems.

#### 2.3.2. Study on Population Structure

There have been successful applications of static population topologies since the early research of particle swarm optimization (PSO), such as [22–24]. In [23,25], basic static population structures, including the star, ring, and von Neumann topologies, and

their influence on the performance of PSO were analyzed. A looser topology generally provides better population diversity, while a tighter topology leads to a faster convergence rate. The population topology mechanism has also been naturally applied to differential evolution [26] and genetic algorithm [27].

Dynamic topologies are common in recent EAs and SIOAs. Some of them are simple extensions of classic static structures. For example, [28] proposed PSO with increasing topology connectivity to adapt to the different needs of different search stages. Other methods achieve greater flexibility by dynamically adapting the topology, including elite strategy [29], clustering [30], or randomization [31].

The hierarchical population topology model applied in this paper can be explained as a combination of basic topology models, sometimes referred to as the island model. Such a framework has already been applied in particle swarm optimization [32], differential evolution [33], genetic algorithm [34], and genetic programming [35]. However, the proposed algorithm adopts covariance matrix adaptation evolution strategy (CMA-ES) for each sub-population and combines the population structure with the geometric relationship in the search space. Therefore, it has a more intuitive and stronger control on the population.

## 3. Hierarchical Collaboration Model

In this section, the theoretical model of the proposed hierarchical framework is built through an information perspective.

For an optimization task, the unknown objective function can be regarded as a random sample from distribution $p(f)$. The desired optimal solution $\mathbf{x}^*$ is then a random variable within feasible space $S$. With history evaluation data $D_t = \{\mathbf{x}_i, y_i\}_{i=1}^t$, the posterior distribution of $f$ can be obtained, so does the posterior of $\mathbf{x}^*$ in Equation (2).

$$p(\mathbf{x}^*|D_t) = \int_f p(f|D_t) \times I\left(\mathbf{x}^* = \arg\min_{x \in S} f(x)\right) \mathrm{d}f \tag{2}$$

With a limited number of sample data, an optimizer targets to obtain information on $\mathbf{x}^*$, that is, to reduce the entropy of its posterior distribution, which is shown in Equation (3).

$$H(p(\mathbf{x}^*|D_t)) = -\int_{\mathbf{x}^* \in S} p(\mathbf{x}^*|D_t) \log p(\mathbf{x}^*|D_t) \mathrm{d}\mathbf{x}^* \tag{3}$$

The expected entropy reduction caused by new data $(\mathbf{x}, y)$ is an ideal evaluation for sampling $\mathbf{x}$ and is adopted in algorithms called entropy search [36]. Consider a partition of the feasible space $\{S_i\}_{i=1}^N$ that $S = \bigcup_{i=1}^N S_i$ and $S_i \cap S_j = \varnothing$, Appendix A indicates that the entropy can be decomposed into each subspace, as in Equation (4).

$$H(p_S(\mathbf{x}^*)) = \sum_{i=1}^N p(\mathbf{x}^* \in S_i) \times H(p_{S_i}(\mathbf{x}^*)) + H(\{p(\mathbf{x}^* \in S_i)\}_{i=1}^N) \tag{4}$$

where condition on history data $D_t$ is abbreviated for each distribution of $\mathbf{x}^*$. $p_S(\mathbf{x}^*)$ means the distribution of optimal solution restricted within $S$. The second term, called region entropy, is the entropy of discrete distribution that describes which local region the global optimal is located.

The entropy decomposition model illustrates that the overall optimization can be performed by the independent optimization within each sub-space $S_i$ and the specification of probability that the global optimal locates in each sub-space. This model corresponds directly to the idea of the fireworks algorithm, in which fireworks optimize in different local areas and collaborate.

The reduction in region entropy in Equation (4) has to be performed by the effective collaboration between fireworks. However, it can also be regarded as a discrete approximation of $H(p_S(\mathbf{x}^*))$ ignoring the local details within each sub-space. When the sub-spaces

become smaller and smaller, the region entropy gradually approximates the global entropy. Inspired from this fact, Equation (4) can be rewritten as follow.

$$
\begin{aligned}
H(p_S(\mathbf{x}^*)) = {}& (1 - \alpha) \times H(p_S(\mathbf{x}^*)) \\
&+ \alpha \times \sum_{i=1}^{N} p(\mathbf{x}^* \in S_i) \times H(p_{S_i}(\mathbf{x}^*)) \\
&+ \alpha \times H(\{p(\mathbf{x}^* \in S_i)\}_{i=1}^{N})
\end{aligned}
\tag{5}
$$

Since the entropy in the third term of Equation (5) is an approximation of entropy in the first term, they can be combined and represented by the entropy of an auxiliary distribution as in Equation (6).

$$
H(p_S(\mathbf{x}^*)) \approx H(p_S^{(\alpha)}(\mathbf{x}^*)) + \alpha \times \sum_{i=1}^{N} p(\mathbf{x}^* \in S_i) \times H(p_{S_i}(\mathbf{x}^*))
\tag{6}
$$

where $p_S^{(\alpha)}(\mathbf{x}^*)$ represents an auxiliary distribution balanced between the original distribution $p_S(\mathbf{x}^*)$ and discrete distribution $p(\mathbf{x}^* \in S_i)$. The reduction in its entropy corresponds to the optimization of a low-fidelity objective function that is smoothed on each sub-region.

Equation (6) describes the theoretical model of the proposed algorithm, which contains $N$ local fireworks optimizing in sub-spaces $\{S_i\}_{i=1}^{N}$ and a global firework optimizing a low-fidelity target in the overall feasible space $S$. Usually, when using a local optimization algorithm for the global firework, its sampling range is a subset of feasible space $S_0 \subset S$ which implies the possible range of the global optimal. In this case, the approximation still holds when $\{S_i\}_{i=1}^{N}$ be a partition of $S_0$.

For any efficient local optimization algorithm, the model illustrates that it is better applied in individual optimization of this framework for the following advantages than direct optimization.

1.  **Task Decomposition.** The hierarchical model decomposes the overall optimization task into $N + 1$ independent optimization tasks. All of those sub-tasks are easier than the original optimization. The linear decomposition equation ensures that more overall entropy reduction can be achieved by processing multiple simpler tasks simultaneously.
2.  **Sample Efficiency.** On a modern parallel computation device, the number of samples that can be evaluated at the same time is much larger than the number of samples needed per generation for most optimization algorithms. By adopting $N + 1$ independent optimization, the proposed framework can utilize the computation device more efficiently.
3.  **Flexibility and Simplicity.** The theoretical model does not make any limitation on the strategy of individual optimization. They can be unified for simplicity or varied for specific requirements. The global firework directly controls the overall balance of exploration and exploitation. No additional collaboration strategy is necessary once the search space partition is satisfied.
4.  **Multi-Scale Optimization.** The framework is particularly suitable for objective functions with both global trends and local patterns, which is quite common in practical problems.

For better geometric intuition, an algorithm adopting a dynamic local Gaussian distribution model for individual optimization is proposed in this paper, which evolves according to an extended CMA-ES algorithm. Furthermore, the collaboration intends to make their search ranges dynamically adjusted towards a partition with the minimal loss of independent optimization efficiency. The advantages will be further examined and discussed in experiments.

## 4. Individual Optimization Strategy

During the optimization, each firework maintains sparks that follow multi-variant Gaussian distributions. The individual optimization strategy samples sparks from the distribution and adapts it according to their evaluations. In the remainder of this section, the unified algorithm flow is first introduced, followed by a detailed explanation of the different parameter choices for local and global fireworks.

### 4.1. Sparks Generation

Sparks of each firework are generated as i.i.d samples of a multi-variant Gaussian distribution according to the following equation.

$$\mathbf{x}_{i,1:\lambda_i} \sim \mathbf{m}_i + \sigma_i \times \mathcal{N}(\mathbf{0}, C_i) \tag{7}$$

where $\lambda_i$ is the number of sparks. $\mathbf{m}_i$ and $C_i$ are the mean and covariance matrix, respectively. $\sigma_i$ is an overall scale factor. All sparks beyond the boundaries are re-mapped with a mirrored mapping rule.

$$\mathbf{x}_{i,j,k} = \begin{cases} 2lb_k - \mathbf{x}_{i,j,k}, & \text{if} \quad \mathbf{x}_{i,j,k} < lb_k \\ \mathbf{x}_{i,j,k}, & \text{if} \quad lb_k \leq \mathbf{x}_{i,j,k} \leq ub_k \\ 2ub_k - \mathbf{x}_{i,j,k}, & \text{if} \quad \mathbf{x}_{i,j,k} > ub_k \end{cases} \tag{8}$$

where $lb_k$ and $ub_k$ are the lower bound and upper bound on the $k$-th dimension, respectively. All the samples are collected and evaluated by the objective function $y_{ij} = f(\mathbf{x}_{ij})$.

### 4.2. Mean Shift

The new mean position $\mathbf{m}_i^{(l)}$ is first adapted as a weighted average of the sparks.

$$\mathbf{m}_i^{(l)} = \mathbf{m}_i + c_m \sum_{j=1}^{\lambda_i} w_{ij}(\mathbf{x}_{ij} - \mathbf{m}_i) \tag{9}$$

where $c_m \in [0, 1]$ is the learning rate. The recombination weights satisfy $w_{ij} \geq 0$ and $\sum_{j=1}^{\lambda_i} w_{ij} = 1$. Usually, the better individuals have higher weights.

### 4.3. Covariance Adaptation

Both rank-$\mu$ update and rank-1 update are applied for the adaptation of covariance matrix according to Equation (10).

$$C_i^{(l)} = (1 - c_\mu - c_1)C_i + c_\mu \sum_{j=1}^{\lambda_i} w_{ij}\mathbf{y}_{ij}\mathbf{y}_{ij}^T + c_1\mathbf{p}_{c,i}\mathbf{p}_{c,i}^T \tag{10}$$

where $c_\mu$ and $c_1$ are learning rates. The second term in the formula corresponds to the rank-$\mu$ update. Sample bias $\mathbf{y}_{ij} = \mathbf{x}_{ij} - \mathbf{m}_i^{(r)}$, where the reference mean $\mathbf{m}_i^{(r)}$ is a position selected between the original and the new mean in order to balance the exploitation and exploration ability.

$$\mathbf{m}_i^{(r)} = (1 - c_r)\mathbf{m}_i + c_r\mathbf{m}_i^{(l)} \tag{11}$$

The third term in Equation (10) corresponds to the rank-1 update, which adjusts the distribution according to the historical trajectory of the mean position. The evolution path $\mathbf{p}_{c,i}$ is updated according to the following equation.

$$\mathbf{p}_{c,i}^{\text{new}} = (1 - c_c)\mathbf{p}_{c,i} + \sqrt{c_c(2 - c_c)\mu_{\text{eff}}} \times \frac{\mathbf{m}_i^{(l)} - \mathbf{m}_i}{\sigma_i} \tag{12}$$

where $c_c$ is the learning rate and $\mu_{\text{eff}} = (\|w\|_1 / \|w\|_2)^2$ is the variance effective selection mass of the sample weights.

### 4.4. Scale Adaptation

The overall scale $\sigma$ is adjusted according to the states of individual optimization, which is illustrated by the conjugate evolution path $\mathbf{p}_\sigma$.

$$\mathbf{p}_{\sigma,i}^{\text{new}} = (1 - c_\sigma)\mathbf{p}_{\sigma,i} + \sqrt{c_\sigma(2 - c_\sigma)\mu_{\text{eff}}} \times C_i^{-\frac{1}{2}} \frac{\mathbf{m}_i^{(l)} - \mathbf{m}_i}{\sigma_i} \tag{13}$$

The Euclidean norm of $\mathbf{p}_\sigma$ is compared with the expectation of sample norm from standard Gaussian. The fact that the conjugate evolution path is longer indicates the mean position has been moving significantly in the same direction, so the overall scale $\sigma$ should be amplified. Otherwise, it will shrink.

$$\ln \sigma_i^{(l)} = \ln \sigma_i + \frac{c_\sigma}{d_\sigma} \left( \frac{\|\mathbf{p}_{\sigma,i}^{\text{new}}\|}{\mathrm{E}\|\mathcal{N}(\mathbf{0}, I)\|} - 1 \right) \tag{14}$$

$c_\sigma$ is the learning rate. Damping factor $d_\sigma$ is used to control the magnitude of the update.

### 4.5. Restart

Fireworks that are stagnant or unlikely to surpass the current optimal should be reset timely. Primary restart conditions include the following.

- **Fitness Converged.** $\mathrm{std}\left[\mathbf{y}_{i,1:\lambda_i}\right] \leq \epsilon_v$.
- **Position Converged.** $\sigma_i \times \|C_i\|_2 \leq \epsilon_p$.
- **Not Improving.** The firework's optimal solution has not improved for $\epsilon_l$ iterations.
- **Mean Converged.** The firework's mean position converges with a better firework, that is, $\|\mathbf{m}_i - \mathbf{m}_j\| < \epsilon_p$.
- **Cover by Better.** The firework's explosion range is covered by a better firework, which is verified when more than 90% of its sparks lie within the explosion range of the other firework.

Once a restart condition is met, the firework is re-initialized in a random location within the feasible space. The explosion range will be defined later in Equation (18).

### 4.6. Parameter Settings

Local and global fireworks require different parameter settings for their distinct targets with the unified individual optimization framework. In general, the local fireworks are desired to conduct an efficient search in local areas. On the other hand, the global firework requires a gradual and steady narrowing of its search space started from the whole feasible space. The parameter settings of both global and local fireworks are described below, and their behaviors are illustrated in Figure 1.

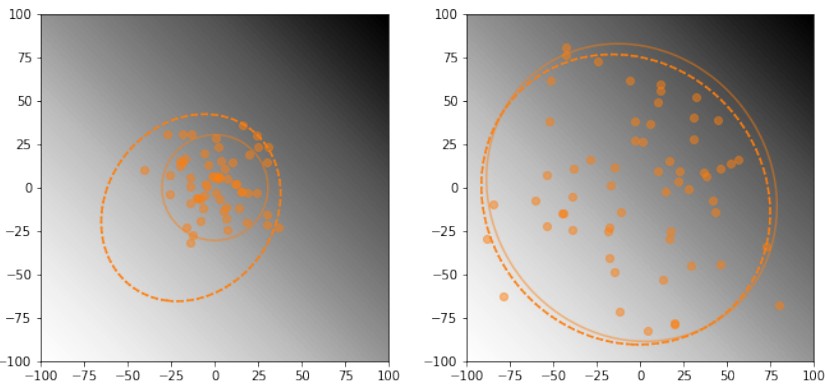

**Figure 1.** Individual optimization of local and global fireworks. The solid and dashed lines are the contours of the distribution before and after the local single-step optimization, respectively.

### 4.6.1. Initialization

Each firework in the proposed algorithm maintains the same number of sparks $\lambda_i = \lfloor \lambda / (N+1) \rfloor$. The local fireworks locate uniformly within the feasible space, while the global firework is sampled near the origin. All covariance matrices are initialized with the identity matrix. Additionally, the overall scale of global firework is shown in Equation (15).

$$\sigma_{global} = \frac{ub - lb}{2 \times E \| \mathcal{N}(\mathbf{0}, I) \|} \tag{15}$$

where $ub = \max(ub_k)$ and $lb = \max(lb_k)$ are the upper bound and lower bound of the feasible space. Local fireworks are initialized with $\sigma_{local} = \sigma_{global} / N$.

### 4.6.2. Recombination Weights

For efficient exploitation, local fireworks select the better half of sparks ($\mu = \lfloor \lambda / 2 \rfloor$) and assign logarithmic weights according to their rank as follows. Here, $w_j$ is the weight for the $j$-th best spark.

$$w_j \propto \min(0, \log(\mu + 0.5) - \log j) \tag{16}$$

For stable exploration, the global firework eliminates 5% worst sparks ($\mu = \lfloor 0.95\lambda \rfloor$) and assign uniform weights for the rest.

### 4.6.3. Referenced Mean

Scalar $c_r \in [0, 1]$ decides the referenced mean and balances the exploration and exploitation of rank-$\mu$ update. Original CMA-ES uses $c_r = 0$ for highly exploratory population. The global firework takes $c_r = 1.0$, so the adapted covariance matrix tends to reproduce its selected samples with maximum probability. The local fireworks take $c_r = 0.5$ for a balanced local optimization.

### 4.6.4. Rank-1 Update

The rank-1 update adjusts the sample distribution based on the historical movement trajectory of the mean position. It is very effective for local fireworks but not suitable for the global firework.

### 4.6.5. Scale Update

The damping factor $d_\sigma$ in Equation (14) control the magnitude of scale update, which is shrunk to balance with the effect of collaboration. The local fireworks reduce $d_\sigma$ to 0.5 times its original value designed in CMA-ES. Although the global takes $d_\sigma = 0$ because it is sufficient to control search range by selection and collaboration.

#### 4.6.6. Restart Conditions

The value and position precisions of all fireworks both take $\epsilon_v = \epsilon_p = 10^{-5}$. Local fireworks restart after $\epsilon_l = 100$ failed iterations, while $N \times \epsilon_l$ is allowed for the global firework.

#### 4.6.7. Overall Learning Rate

The converge rate of the global firework is already limited through parameters selection. However, during the optimization, it is desired to be significantly slower than local fireworks. An additional overall learning rate $c_g$ is set for the global firework to slow down its optimization further as follows. $c_g = 1/N$ is assigned in the proposed algorithm.

$$
\begin{aligned}
\mathbf{m}_0^{(l)} &\leftarrow c_g \mathbf{m}_0^{(l)} + (1 - c_g) \times \mathbf{m}_0 \\
C_0^{(l)} &\leftarrow c_g C_0^{(l)} + (1 - c_g) \times C_0 \\
\sigma_0^{(l)} &\leftarrow c_g \sigma_0^{(l)} + (1 - c_g) \times \sigma_0
\end{aligned}
\tag{17}
$$

### 4.7. Individual Optimization Framework

The framework of individual optimization strategy is shown in Algorithm 1.

---
**Algorithm 1** Individual optimization framework.

---
**for all** firework $F_i$ **do**
    Generate sparks $\mathbf{x}_{ij}$ according to Equation (7)
**end for**
Gather and evaluate sparks $y_{ij} = f(\mathbf{x}_{ij})$
**for all** firework $F_i$ **do**
    Compute $\mathbf{m}_i^{(l)}$ according to Equation (9)
    Update evolution path $\mathbf{p}_i$ according to Equation (12)
    Compute $C_i^{(l)}$ according to Equation (10)
    Update conjugate evolution path $\mathbf{p}_{c,i}$ according to Equation (13)
    Compute overall scale $\sigma_i^{(l)}$ according to Equation (14)
**end for**
Adjust global firework according to Equation (17)

---

## 5. Collaboration Strategy

Based on the model in Equation (6), the proposed algorithm mainly considers collaboration by adjusting fireworks' search ranges towards a partition. Since each firework maintains a multi-variant Gaussian distribution, its search space is defined by a bounded region where the probability density exceeds a specific value.

$$
B_F = \left\{ \mathbf{x} \,\big|\, \|C^{-\frac{1}{2}}(\mathbf{x} - \mathbf{m})/\sigma\| = d_B \right\}
\tag{18}
$$

Equation (18) defines the boundary of firework $F_i$ as $B_{F_i}$, abbreviated as $B_i$, which corresponds to an ellipsoidal shell. Then its closure $S_F = \overline{B}_F$ is the firework's search range. In the proposed algorithm, $d_B = \text{mean}(\chi_D) + 0.5 \times \text{std}(\chi_D)$ is taken, by which the defined search range $S_F$ covers about 70% of samples for arbitrary dimension $D$.

For simplicity, the collaboration of firework's search range is approximated in pairs, including the following steps.

### 5.1. Computation of Dividing Points

For each pair of fireworks $F_i$ and $F_j$, a dividing point can be obtained on the connecting line $\mathbf{m}_i\mathbf{m}_j$, which specifies the cut-off point of their boundaries under ideal collaboration. Let the radius of $S_i$ on the line $\mathbf{m}_i\mathbf{m}_j$ be $r_{ij}$ and $d_{ij} = \|\mathbf{m}_i\mathbf{m}_j\|$. The dividing point $\mathbf{d}_{ij}$ can

be obtain by adjusting $r_{ij}$ and $r_{ji}$ simultaneously until their boundaries coincide on $\mathbf{m}_i\mathbf{m}_j$. For local fireworks, this is completed by solving the following equation.

$$r_{ij}e^{a_{ij}w} + r_{ji}e^{a_{ji}w} = d_{ij} \tag{19}$$

where $r_{ij}$ changes to $r_{ij}e^{a_{ij}w}$ by collaboration, which also equals to $\|\mathbf{m}_i\mathbf{d}_{ij}\|$. The sensitivity factors $a_{ij}$ and $a_{ji}$ control the relative magnitudes of change for $F_i$ and $F_j$. The collaboration is also presented in Figure 2.

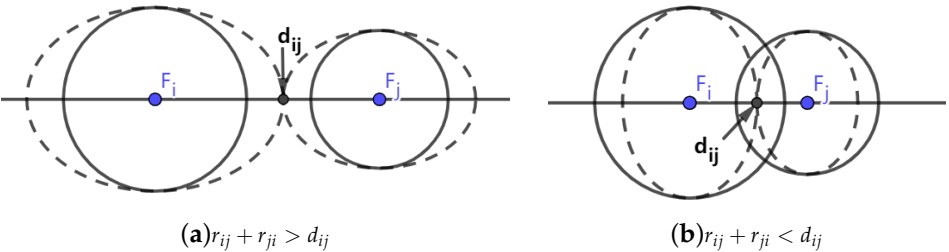

(**a**)$r_{ij} + r_{ji} > d_{ij}$          (**b**)$r_{ij} + r_{ji} < d_{ij}$

**Figure 2.** Dividing Point in local collaboration.

The following equation should be solved when $F_i$ is the global firework.

$$r_{ij}e^{-a_{ij}w} - r_{ji}e^{a_{ji}w} = d_{ij} \tag{20}$$

where the negative sensitivity of global firework indicates that it changes in the opposite direction to local firework. The collaboration with global firework is presented in Figure 3.

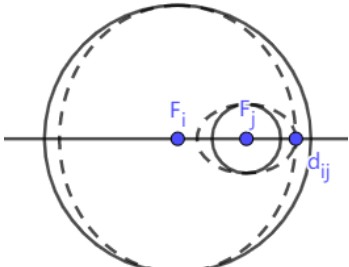

**Figure 3.** Dividing point in global collaboration.

The sensitivity factors are set to balance the influence of global collaboration on individual optimizations. A comparison is made between their optimization states for each pair of fireworks $F_i$ and $F_j$. Only when the worst spark of $F_i$ is still better than the best of $F_j$, let $a_{ij} = 0.0$ and $a_{ji} = 1.0$ so the optimization of $F_i$ will not be influenced. Otherwise, let $a_{ij} = a_{ji} = 1.0$ so they collaborate by the same magnitude. In order to protect effective individual optimization, the sensitivity of a local firework is adjusted to zero if it improves within $0.2\epsilon_l$ iterations. When both sensitivities are zeros, dividing points fall on the boundary of each of the fireworks, so they do not change in collaboration.

The sensitivity factor of global firework is amplified by $c_a > 1.0$ to ensure the efficiency of local optimization. In the proposed algorithm, a significant large value $c_a = 5.0$ is taken.

For both equations, the value on the left is always monotonic to $w$ when the sensitivity factors are positive. Therefore, it is guaranteed to be solved quickly for a given accuracy and range.

### 5.2. Selection of Feature Points

The dividing points describe the boundaries for each pair of fireworks under the complete and ideal collaboration. However, only several dividing points should be selected and adjusted for firework $F_i$ during optimization.

First, the collaboration should only be performed between adjacent fireworks. In order to avoid the complex computation of geometric relationship in high-dimensional space, the $\tau = 2$ most critical points $\{\mathbf{d}_{ik}\}_{k=1}^{\tau}$ are selected for each firework. For local fireworks, dividing points with the highest probability density are selected. On the contrary, dividing points with the lowest density are selected for the global firework.

Then, the selected points should be clipped towards the distribution boundary to avoid excessive change. Based on the analysis of individual searches, a similar magnitude of change was taken within $[0.85, 1.20]$. The feature point is calculated in Equation (21), and the clipping is shown in Equation (22).

$$\mathbf{f}_{ik} = \mathbf{m}_i^{(l)} + d_{ik}^{(clip)} \times \frac{\mathbf{d}_{ik} - \mathbf{m}_i^{(l)}}{\left\| \mathbf{d}_{ik} - \mathbf{m}_i^{(l)} \right\|} \tag{21}$$

where $d_{ik}^{(clip)}$ is the distance from mean position $\mathbf{m}_i^{(l)}$ to feature point $\mathbf{f}_{ik}$.

$$d_{ik}^{(clip)} = \begin{cases} \alpha_l r_{ij}, & if \left\| \mathbf{d}_{ik} - \mathbf{m}_i^{(l)} \right\| < 0.85 r_{ij} \\ \alpha_u r_{ij}, & if \left\| \mathbf{d}_{ik} - \mathbf{m}_i^{(l)} \right\| < 1.20 r_{ij} \\ \left\| \mathbf{d}_{ik} - \mathbf{m}_i^{(l)} \right\|, & else. \end{cases} \tag{22}$$

### 5.3. Adaptation to Feature Points

The distribution of $F_i$ after individual adaptation is then further adjusted to fit the feature points $\{\mathbf{f}_{ik}\}_{k=1}^{\tau}$ on its boundary. For simplicity, the fitting is performed one by one for each feature point.

### 5.3.1. Mean Shift

First, the mean position is shifted to balance the overall boundary shape change. Then, for each feature point $\mathbf{f}_{ik}$, the reverse of its shortest vector to the boundary $B_i$ is averaged. The shift vector of the mean position is calculated as follows.

$$\mathbf{mv}_i = \frac{1}{\tau} \sum_{k=1}^{\tau} \left( \mathbf{f}_{ik} - \mathbf{q}_{ik} \right) \tag{23}$$

The point $\mathbf{q}_{ik}$ is the closer intersection of $B_i$ and line $\mathbf{m}_i \mathbf{f}_{ik}$. The shifting vector is restricted within 20% of the radius on the corresponding direction for local fireworks and 5% for the global firework.

$$\mathbf{m}_i^{(g)} = \mathbf{m}_i^{(l)} + \mathbf{mv}_i \times \min \left\{ 1, \frac{\alpha_m r_i}{\|\mathbf{mv}_i\|} \right\} \tag{24}$$

According to experiments, the mean shift is also helpful in reducing the condition number of the resulted covariance matrix and avoiding possible numerical problems.

### 5.3.2. Boundary Adaptation

The boundary collaboration is completed separately for each feature point and averaged. It is easy to examine the following theorem by simply substituting the matrix in Equation (25) into the definition of firework boundary.

**Theorem 1.** *For a multi-variant Gaussian distribution $\mathcal{N}(\mathbf{m}^{(g)}, C^{(l)})$ with overall scale $\sigma^{(l)}$ and a feature point $\mathbf{f}$, the following matrix $C_{\mathbf{f}}^{(g)}$ satisfies that $\mathbf{f}$ is on its boundary and it has the same radius as $C^{(l)}$ on the conjugate directions.*

$$C_{\mathbf{f}}^{(g)} = C^{(l)} + \frac{\lambda}{\sigma^2} \times (\mathbf{f} - \mathbf{m}^{(g)})(\mathbf{f} - \mathbf{m}^{(g)})^T \tag{25}$$

*where*

$$\lambda = \frac{1}{d_B^2} - \frac{1}{\mathbf{z}^T \mathbf{z}} \tag{26}$$

*and*

$$\mathbf{z}_{ik} = \left( C_i^{(l)} \right)^{-\frac{1}{2}} \times (\mathbf{f}_{ik} - \mathbf{m}_i^{(g)}) / \sigma_i^{(l)} \tag{27}$$

Then, the collaborated covariance matrix $C_i^{(g)}$ is obtained by averaging the effects of each feature point.

$$C_i^{(g)} = C_i^{(l)} + \frac{1}{\tau} \sum_{k=1}^{\tau} \frac{\lambda_{ik}}{\sigma_i^2} \times (\mathbf{f}_{ik} - \mathbf{m}_i^{(g)})(\mathbf{f}_{ik} - \mathbf{m}_i^{(g)})^T \tag{28}$$

The distribution scale is changed along with the covariance matrix, so $\sigma_i$ remains the same.

$$\sigma_i^{(g)} = \sigma_i^{(l)} \tag{29}$$

### 5.4. Collaboration Framework

The proposed algorithm is shown in Algorithm 2. An overall reboot is conducted when the population has not improved for over $M = 100$ iterations.

---

**Algorithm 2** Hierarchical collaborated fireworks algorithm.

---

**while** The termination condition is not satisfied **do**
    Initialize local fireworks population $\{F_i\}_{i=1}^n$
    Initialize global firework $F_0$
    **while** the population improved within $M$ iterations **do**
        # Individual Optimization
        **for all** firework $F_i, i = 0, 1, ..., n$ **do**
            Generate sparks $\mathbf{x}_{ij}$
        **end for**
        Collect and evaluate all sparks
        **for all** firework $F_i, i = 0, 1, ..., n$ **do**
            Compute $\mathbf{m}_i^{(l)}, C_i^{(l)}, \sigma_i^{(l)}$
            **if** any restart condition is satisfied **then**
                Re-initialize $F_i$
            **end if**
        **end for**
        # Collaboration
        **for all** pair of fireworks $F_i$ and $F_j$ **do**
            compute dividing point $\mathbf{d}_{ij}$ as in Section 5.1
        **end for**
        **for all** firework $F_i$ **do**
            select and adjust feature points $\{\mathbf{f}_{ik}\}_{k=1}^{\tau}$ as in Section 5.2
            compute $\mathbf{m}_i^{(g)}, C_i^{(g)}, \sigma_i^{(g)}$ as in Section 5.3
            update firework distribution: $\mathbf{m}_i \leftarrow \mathbf{m}_i^{(g)}, C_i \leftarrow C_i^{(g)}, \sigma_i \leftarrow \sigma_i^{(g)}$
        **end for**
    **end while**
**end while**
Return the best evaluated solution

---

## 6. Experiments and Discussions

In this section, the efficiency and properties of the proposed algorithm are experimented and discussed. All the experiments are run on Ubuntu 18.04 with Intel(R) Xeon(R) CPU E5-2675 v3.

### 6.1. Experiments on the Algorithm Efficiency

The efficiency of the proposed algorithm is examined on the bound-constrained single objective benchmark test suite from the IEEE Congress on Evolutionary Computation (CEC) 2020 [16]. The benchmark set contains ten objective functions, including uni-modal, basic, hybrid, and composition problems. It provides a relatively comprehensive test of different aspects of performance for optimization algorithms. Since consistent experimental results are observed in 10, 15, and 20 dimensions, only 20-dimensional performance is presented in this paper.

The hierarchical collaborated FWA (HCFWA) is first compared with previous important FWAs to examine the effectiveness of the proposed strategy. The loser-out tournament FWA (LoTFWA [15]) is a classical FWA that has received widespread attention and served as the foundation of many subsequent studies. CMAFWA replaces the local search of LoTFWA with CMA-ES and is used as a comparison algorithm. FWA based on search space partition (FWASSP [3]) uses a similar collaboration strategy with the proposed algorithm but only adopts local fireworks.

Each algorithm is tested 30 times on 20-dimensional problems with a maximum number of 10,000,000 evaluations. The proposed algorithm keeps most parameters consistent with the previous ones as they were published. The results of the longitudinal comparisons are shown in Table 1. The best results for each problem are shown in bold.

**Table 1.** Comparison with FWAs on 20 dimensional problems of CEC 2020.

| | LoTFWA | | | CMA-FWA | | | FWASSP | | | HCFWA | |
|---|---|---|---|---|---|---|---|---|---|---|---|
| Func. | Mean | Std | | Mean | Std | | Mean | Std | | Mean | Std |
| 1 | 1.625e+06 | 4.048e+05 | + | **0.000e+00** | 0.000e+00 | − | 1.238e–05 | 3.640e–06 | = | 1.751e–05 | 1.929e–06 |
| 2 | 1.531e+03 | 4.151e+02 | + | **2.647e+02** | 1.215e+02 | − | 4.299e+02 | 1.681e+02 | = | 6.815e+02 | 2.293e+02 |
| 3 | 6.873e+01 | 9.701e+00 | + | 2.437e+01 | 8.288e–1 | + | 6.181e+02 | 2.962e+01 | + | **1.721e+01** | 8.254e+00 |
| 4 | 1.074e+01 | 1.604e+00 | + | 1.421e+00 | 3.200e–1 | + | 1.867e+00 | 6.521e–01 | + | **6.636e–01** | 1.049e–01 |
| 5 | 2.692e+05 | 1.768e+05 | + | 1.230e+03 | 3.018e+02 | + | **1.891e+02** | 4.939e+01 | − | 3.757e+02 | 1.054e+02 |
| 6 | 4.579e+02 | 2.063e+02 | + | 7.375e+00 | 7.963e+00 | + | 1.594e+02 | 5.865e+01 | + | **1.632e+00** | 2.585e–01 |
| 7 | 6.508e+04 | 5.798e+04 | + | 4.565e+02 | 2.158e+02 | + | **1.005e+02** | 4.884e+01 | − | 2.374e+02 | 8.577e+01 |
| 8 | 1.084e+02 | 1.010e+01 | + | 4.589e+02 | 1.463e+02 | + | 1.000e+02 | 2.272e–07 | + | **7.201e+01** | 4.043e+01 |
| 9 | 4.505e+02 | 1.856e+01 | + | 4.049e+02 | 1.659e+00 | + | 2.112e+02 | 9.651e+01 | + | **1.067e+02** | 2.494e+01 |
| 10 | 4.185e+02 | 1.358e+01 | + | 4.063e+02 | 5.418e–03 | − | **4.024e+02** | 5.840e+00 | = | 4.028e+02 | 5.095e+00 |
| Result | 10 vs. 0 | | | 8 vs. 2 | | | 5 vs. 2 | | | | |
| AR | 3.80 | | | 2.40 | | | 2.10 | | | **1.70** | |

The Wilcoxon rank-sum tests are performed to verify the difference in optimization results' significance. The proposed algorithm obtained the best average results on five problems. CMAFWA presents the most outstanding local optimization ability with the original CMA-ES strategy, especially on the first and second problems: a uni-modal problem and a multi-modal problem with high local search capability requirements. FWASSP and HCFWA, with fireworks collaboration, obtain all the best average results on the rest problems. However, their local exploitation ability is slightly weaker because of the defined minimum optimization precision $\epsilon_v = \epsilon_p = 10^{-5}$ and the adverse effects of collaboration. With a confidence level of 95%, HCFWA outperforms FWASSP significantly on five problems but is worse on two problems, which might be caused by an improper global convergence rate.

The proposed algorithm is also compared with important algorithms, including IPOP-CMA-ES [37] and SHADE [38], which are both currently the most widely applied and efficient evolutionary algorithms. The results of LoTFWA, IPOP-CMA-ES, SHADE, and HCFWA are listed in Table 2. It has been complicated for firework algorithms to complete with various differential evolution (DE) algorithms on benchmarks after CEC

2017. The proposed algorithm achieved similar results with SHADE and is better suited to specific problems.

**Table 2.** Comparison with classic algorithms on 20 dimensional problems of CEC 2020.

| | LoTFWA | | | IPOP-CMA-ES | | | SHADE | | | HCFWA | |
|---|---|---|---|---|---|---|---|---|---|---|---|
| **Func.** | **Mean** | **Std** | | **Mean** | **Std** | | **Mean** | **Std** | | **Mean** | **Std** |
| 1 | 1.63e+06 | 4.05e+05 | + | **0.00e+00** | 0.00e+00 | − | **0.00e+00** | 0.00e+00 | − | 1.75e−05 | 1.93e−06 |
| 2 | 1.53e+03 | 4.15e+02 | + | 2.16e+03 | 2.41e+01 | + | **2.16e+01** | 9.14e+00 | − | 6.82e+02 | 2.29e+02 |
| 3 | 6.87e+01 | 9.70e+00 | + | 5.43e+01 | 7.97e+00 | + | 2.08e+01 | 2.20e−01 | = | **1.72e+01** | 8.25e+00 |
| 4 | 1.07e+01 | 1.60e+00 | + | 2.32e+00 | 2.78e−01 | + | 6.48e−01 | 6.49e−02 | = | 6.64e−01 | 1.05e−01 |
| 5 | 2.69e+05 | 1.77e+05 | + | 1.23e+03 | 2.83e+02 | + | **4.37e+01** | 3.89e+01 | − | 3.76e+02 | 1.05e+02 |
| 6 | 4.58e+02 | 2.06e+02 | + | 4.91e+02 | 2.19e+00 | + | 2.07e+00 | 2.12e−01 | + | **1.63e+00** | 2.59e−01 |
| 7 | 6.51e+04 | 5.80e+04 | + | 7.18e+02 | 2.10e+02 | + | **1.50e+00** | 9.57e−01 | − | 2.37e+02 | 8.58e+01 |
| 8 | 1.08e+02 | 1.01e+01 | + | 2.48e+03 | 1.85e+02 | + | 1.00e+02 | 0.00e+00 | + | **7.20e+01** | 4.04e+01 |
| 9 | 4.51e+02 | 1.86e+01 | + | 4.32e+02 | 1.48e+00 | + | 4.07e+02 | 2.19e+00 | + | **1.07e+02** | 2.49e+01 |
| 10 | 4.19e+02 | 1.36e+01 | + | 4.30e+02 | 4.55e−01 | + | 4.06e+02 | 6.97e−03 | + | **4.03e+02** | 5.10e+00 |
| Result | 10 vs. 0 | | | 9 vs. 1 | | | 4 vs. 4 | | | | |
| AR | 3.60 | | | 3.25 | | | **1.55** | | | 1.60 | |

Although it remains tough to compete with some state-of-art implementations of DE such as [17,39] on this benchmark, HCFWA achieves better results on problems with a large number of local optimal, such as function (3) and function (8), which will be further discussed in later experiments. Function (2) also contains plenty of local optimal, but many local areas have huge condition numbers, which is disadvantageous to the proposed algorithm's individual optimization.

### 6.2. Experiments on the Population Behavior

In the second set of experiments, the effect of global firework is analyzed and examined from the perspective of population behavior.

The population behavior of the fireworks algorithm is quite different compared with traditional evolutionary algorithms. For most EAs, such as CMA-ES or DE, their populations are first dispersed throughout the feasible space and then gradually converge to a single point. However, fireworks in FWA are uniformly distributed within the feasible space, and each sub-population converges to its corresponding firework. As a result, the population of EAs eventually converges even for multi-modal problems, while the population of the fireworks algorithm usually does not.

The proposed algorithm forms a compromise between those two types of methods. The optimization of global firework makes the overall population distribution converge gradually such as a general evolutionary algorithm. Meanwhile, the local fireworks remain relatively independent and rapidly exploit their local areas. The 2D visualizations of the optimization progress are presented in Figure 4 to show the behavior of the population in the proposed algorithm.

Images in the first row present the optimization process on the uni-modal cigar function. As all fireworks approach the exact global optimum, the search ranges of local fireworks quickly connect. The global firework also narrows its boundary quickly under both individual optimization and collaboration. Then, the collaboration prevents local fireworks' search ranges from overlapping. Only the best local firework independently exploits around the optimum position and converges. The global firework gradually narrows down the distribution of the whole population at a much slower rate. Moreover, the rest local fireworks fill the remained search area of the global fireworks.

Images in the second row present the optimization process on the multi-modal Rastrigin's function. In this case, each local firework might converge to a local minimum near

its initial position. Only when some local firework is not able to keep similar progress of optimization with a better neighbor, it becomes more sensitive in collaboration and fills up the search range of the global firework. In the last picture, two local fireworks converged, while the other two are much more collaborative. The global firework keeps slowly narrowing around the local fireworks. This steady-state will be maintained until a converged firework finishes its local exploitation or another firework fails to discover potential solutions within the remained space of global firework for too long.

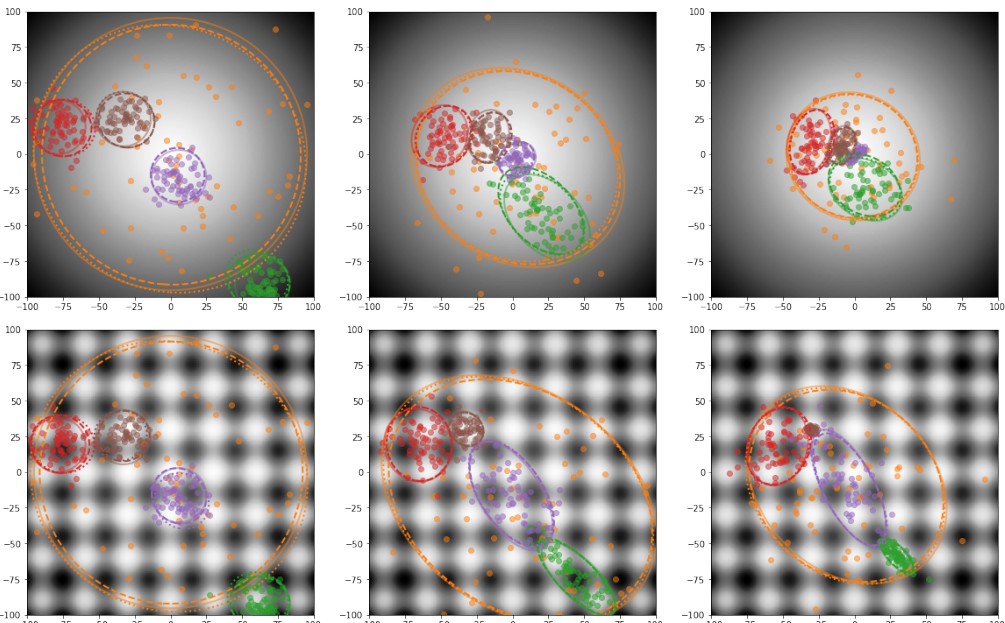

**Figure 4.** Optimization process of HCFWA on 2D Functions. The solid, dashed, and dotted eclipses correspond to each firework's original distribution, locally adapted distribution, and collaborated distribution. The dots represent sparks. The learning rates of fireworks' individual optimization are reduced for better visualization.

*6.3. Experiments on the Algorithm Applicability*

Combining the analysis on theoretical modal and population behavior, the applicability of the proposed algorithm is discussed and examined below.

According to the "no free lunch" (NFL) theorem [40], any optimization algorithm can only be effective for a particular class of target problems. The distinct population behaviors of classic EAs and FWAs result in different performance characteristics during optimization, leading to their respective suitability for objective functions.

Evolutionary algorithms usually widely distribute their population within the feasible space and slowly narrow the search range until convergence. Therefore, it always considers the global trend of objective functions first and turns to the local pattern after entering a specific local area. On the other hand, the fireworks algorithm directly starts exploiting random local areas. Therefore, the global trend information needs to be obtained and utilized by an effective collaboration strategy, usually absent from the previous algorithms. As analyzed before, the proposed HCFWA combines both types of population behaviors. The global firework keeps continuous optimization on the global trend. Meanwhile, the local fireworks keep exploitation of different local areas. This compromise is built on collaboration and achieved at the cost of the individual optimization efficiency of some worse fireworks.

A set of toy experiments on Rastrigin's function are conducted to show the optimization characteristics of those algorithms. CMAFWA, SHADE, and HCFWA with different global firework learning rates are tested on the adjusted objective function, containing many local minimums and considerable high-frequency amplitude. Each algorithm is repeated 30 times. The mean and variance value of the best fitness evaluated in each iteration are present in Figure 5.

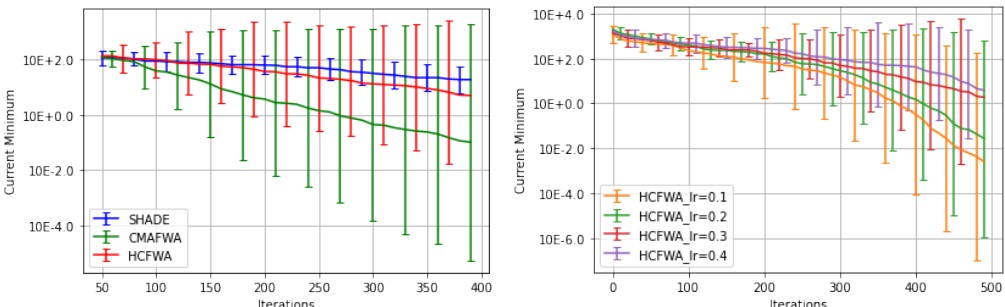

**Figure 5.** Optimization curves on the Rastrigin's function.

The first image in Figure 5 presents the optimization performances of CMAFWA, SHADE, and HCFWA. CMAFWA, and SHADE exhibit very different search patterns. As discussed before, the single population evolutionary algorithm SHADE first observes the global trend of the objective function. Therefore, it improves relatively slowly but is stable with significantly low variance. On the contrary, CMAFWA directly exploits random local areas, improving rapidly in the early stage and having a considerable variance. As HCFWA shares properties of both algorithms, its performance also lies in between.

The second image in Figure 5 presents the optimization performance of HCFWA with different global firework learning rates. As the learning rate gradually increases, the behavior pattern of the proposed algorithm gradually shifts from CMAFWA to SHADE. However, if the learning rate is too large, the pattern can become much more complicated due to the restart of the global fireworks.

Such an optimization pattern of the proposed algorithm indicates that it is more suitable for problems that have both global trends and local patterns. Its convergence rate is better specified based on the global structure of the target problem. When the global feature is significant, the global firework should optimize faster and guide the local fireworks to more potential areas. When the global feature is not significant, the global firework should optimize slowly or stop to guide the local fireworks to broad exploration. Fortunately, such a global structure can usually be evaluated at a relatively small cost in plenty of practical optimization scenarios by techniques, such as approximations, sampling, and simulations.

## 7. Conclusions

This paper extends the fireworks algorithm based on search space partition into a hierarchical collaborative framework. A theoretical model is developed from an information-theoretic perspective and used to guide the algorithm's design and analyze its properties. In the proposed framework, multiple local fireworks exploit each local area, such as classic fireworks algorithms, while a global firework optimizes on a larger scale and controls the overall distribution of the whole population. The hierarchical collaborated fireworks algorithm is implemented based on a unified individual optimization algorithm and collaboration strategy. Experimental results on the CEC 2020 benchmark demonstrate that the proposed algorithm achieved better performance than former variants of FWA and a competitive efficiency compared with other successful frameworks, especially for complex multi-modal problems. Additional experiments are provided to analyze the properties of the proposed framework. It can be observed that HCFWA can simultaneously maintain optimizations on the global trend and local patterns at multiple locations. Therefore, the stability of global exploration and the convergence speed of local exploitation can be guaranteed simultaneously.

Based on the theoretical model of hierarchical collaborative fireworks algorithm, it can analyze the fundamental principle of multi-local and multi-scale optimization and helps build effective optimization algorithms with multiple populations. Furthermore, the significantly experimental results also imply the outstanding ability of the proposed algorithm on specific types of problems. The fireworks algorithm based on such a model contains the considerable potential for further efficiency improvements.

The proposed hierarchical collaborated fireworks algorithm is a basic implementation of a theoretical modal. There are still tremendous possible improvements in many approximation details in collaboration. For example, the utilization of the spatial neighbor relationship of fireworks and the dynamic strategy adjustment of global fireworks seem to have significant effects on the efficiency improvement of the algorithm.

**Author Contributions:** Conceptualization, Y.L.; Data curation, Y.L.; Formal analysis, Y.L.; Funding acquisition, Y.T.; Investigation, Y.L.; Methodology, Y.L.; Project administration, Y.T.; Resources, Y.T.; Software, Y.L.; Supervision, Y.T.; Validation, Y.L.; Visualization, Y.L.; Writing—original draft, Y.L.; Writing—review and editing, Y.T. All authors have read and agreed to the published version of the manuscript.

**Funding:** This work is supported by the National Natural Science Foundation of China (Grant No. 62076010), and partially supported by Science and Technology Innovation 2030—"New Generation Artificial Intelligence" Major Project (Grant Nos.: 2018AAA0102301 and 2018AAA0100302). (Y.T. is the corresponding author.)

**Institutional Review Board Statement:** Not applicable.

**Informed Consent Statement:** Not applicable.

**Data Availability Statement:** Not applicable.

**Conflicts of Interest:** The authors declare no conflicts of interest.

## Appendix A. Decomposition of Entropy

Here, the entropy decomposition of random variable $x \in S$ into the partition $\{S_i\}_{i=1}^{n}$ is presented in Equation (A1). During the derivation, $p(x \in A) = p(x)/p(A)$ presents the distribution restricted in $A \subset S$. Additionally, $H(x \in A)$ corresponds to the entropy of $p(x \in A)$.

$$
\begin{aligned}
H(x \in S) &= -\int_{x \in S} p(x \in S) \log p(x \in S) \mathrm{d}x \\
&= -\sum_{i=1}^{n} \int_{x \in S_i} p(x \in S) \log p(x \in S) \mathrm{d}x \\
&= -\sum_{i=1}^{n} \int_{x \in S_i} p(x \in S_i) p(S_i) \log p(x \in S_i) p(S_i) \mathrm{d}x \\
&= -\sum_{i=1}^{n} \int_{x \in S_i} p(x \in S_i) p(S_i) \log p(x \in S_i) \mathrm{d}x \\
&\quad - \sum_{i=1}^{n} \int_{x \in S_i} p(x \in S_i) p(S_i) \log p(S_i) \mathrm{d}x \\
&= \sum_{i=1}^{n} p(S_i) H(x \in S_i) + H(p(S_i))
\end{aligned} \tag{A1}
$$

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
