# Peer review of "Hierarchical Collaborated Fireworks Algorithm"

_electronics, doi:10.3390/electronics11060948_

Round 1
Reviewer 1 Report
In this paper, the authors extend the theoretical model of fireworks algorithm based on search space partition to obtain a hierarchical collaboration model. Experimental results validate that the hierarchical collaborated fireworks algorithm outperforms former fireworks algorithms. 1. Abstract is well written 2. Pl state the novelty of the work in Introduction. 3. Can Fireworks algorithm be explained with a short example? 4. Related work is shallow. Can include some state-of-the-art works such as: A Soft Computing based Multi-Objective Optimization Approach for Automatic Prediction of Software Cost Models The Combined Study of Improved Fuzzy Optimization Techniques with the Analysis of the Upgraded Facility Location Center for the Covid-19 Vaccine by Fuzzy Clustering Algorithms 5. Equations should be explained in the text and should be cited. 6. Algorithms 1 and 2 should be clearly described. 7. Pl represent the experimental setup as a table. 8. What is the outcome of the proposed work in different scenario? What are the limitations of the work? 9. Compare the approach with existing recent works. 10. Conclusion should conclude the paper and not summarize it.
Author Response
We are really grateful for your review. There are our responses to your comments.
- Thanks!
- The novelty of the work is introduced at the end of Section.I, where we explained as four major contributions of the work.
- We added a short explanation of fireworks algorithm progress in Section2.2.
- We extended the related works in Section.2.3.
- We ensured that each equation is explained in its context or referred to.
- We ensured that each step in the algorithms is clear or referred to a clear equation or description.
- The experiments are simply repeated runs of optimization on each objective function. There are little details on experiment setups as a table.
- The outcome of the proposed work in different scenarios and limitations of the work are discussed in Section.6.3, in which we experimented and analyzed the applicability of the proposed algorithm.
- The comparison is provided in Section.6.1
- Additional conclusion paragraph are provided in Section.7.
Reviewer 2 Report
The paper has some grammatical mistake and spelling mistakes. Need a proof read of the whole paper.
Author Response
We are very grateful for your review. The paper is carefully checked and corrected.
Reviewer 3 Report
The work extends the firework algorithms by applying multiple simultaneous local searches. In particular, the authors develop the theoretical model of firework algorithms based on hierarchical collaboration methods. The related experiments on CEC 2020 verify the effectiveness of the proposed algorithms.
Author Response
We are very grateful for your review.
Reviewer 4 Report
The research aim has been approached systematically. However, below are some notes that you need to consider to make the article easy to follow.
- Some of the used symbols are not defined priorly.
- There is no need for the appendix; its contents could be embedded with the manuscript.
- There are many typos and grammatical errors in the manuscript; thus, it is strongly recommended that the whole work be proofread carefully.
Author Response
We are very grateful for your review.
1. We checked and corrected the use of symbols in the revised manuscript.
2. The inference process is not relevant to the context, so it is arranged in the appendix. It also helps with the typesetting.
3. The paper is carefully checked and corrected.
Round 2
Reviewer 1 Report
All comments are addressed.